# Contributions of Multiple Built Environment Features to 10-Year Change in Body Mass Index and Waist Circumference in a South Australian Middle-Aged Cohort

**DOI:** 10.3390/ijerph17030870

**Published:** 2020-01-30

**Authors:** Suzanne J. Carroll, Michael J. Dale, Anne W. Taylor, Mark Daniel

**Affiliations:** 1Health Research Institute, University of Canberra, Bruce, ACT 2617, Australia; michael.dale@canberra.edu.au (M.J.D.); mark.daniel@canberra.edu.au (M.D.); 2Discipline of Medicine, The University of Adelaide, Adelaide, SA 5000, Australia; anne.taylor@adelaide.edu.au; 3Department of Medicine, St. Vincent’s Hospital, The University of Melbourne, Fitzroy, VIC 3065, Australia

**Keywords:** built environment, physical activity environment, food environment, longitudinal, body mass index, waist circumference, area socioeconomic status

## Abstract

Residential areas may shape health, yet few studies are longitudinal or concurrently test relationships between multiple residential features and health. This longitudinal study concurrently assessed the contributions of multiple environmental features to 10-year change in clinically measured body mass index (BMI) and waist circumference (WC). Longitudinal data for adults (18+ years of age, n = 2253) from the north-west of Adelaide, Australia were linked to built environment measures representing the physical activity and food environment (expressed for residence-based 1600 m road-network buffers) and area education. Associations were concurrently estimated using latent growth models. In models including all environmental exposure measures, area education was associated with change in BMI and WC (protective effects). Dwelling density was associated with worsening BMI and WC but also highly correlated with area education and moderately correlated with count of fast food outlets. Public open space (POS) area was associated with worsening WC. Intersection density, land use mix, greenness, and a retail food environment index were not associated with change in BMI or WC. This study found greater dwelling density and POS area exacerbated increases in BMI and WC. Greater area education was protective against worsening body size. Interventions should consider dwelling density and POS, and target areas with low SES.

## 1. Introduction

Excess body weight as a function of excess adiposity (not muscularity or pregnancy) is broadly accepted as a leading threat to health worldwide and is a major risk factor for cardiometabolic disease [1]. Obesity has reached epidemic proportion (particularly, but not only in developed nations) and the difficulty in addressing this issue is highlighted by the World Health Organisation’s current objective being merely to halt the continuing rise in obesity prevalence [2].

Prevention of overweight/obesity has largely focused on changing individual-level behavioural risk factors such as diet and physical activity [3]. Such strategies have had limited success, with the durability of any resultant behavioural changes in doubt [4]. There is now a push for public health initiatives to include attention to environmental and policy factors that may inhibit or enable healthful behaviours and lifestyles [5]. It is therefore important to identify which modifiable features of environments most strongly impact population health outcomes.

Associations between low area socioeconomic status (SES) and adverse health outcomes are well established [6,7]. Review articles report generally consistent inverse associations between area SES and overweight/obesity, associations which are robust to the inclusion of individual-level factors e.g., [6]. Though area SES may not itself be particularly modifiable, area SES tends to covary with features of the built environment, for example, fast food availability [7]. These built environment features may provide opportunities for health intervention through strategic changes in local policy and urban planning.

The built environment, the human-made or modified space in which people live out their lives [8], may shape health by enabling or constraining health-related behaviours such as diet and physical activity through the availability of, and accessibility to, relevant resources [9]. Access to, and availability of fast food have been linked to unhealthful diet, obesity and type 2 diabetes mellitus (T2DM) [10,11,12]. Conversely, access to and availability of healthful foods are linked with a healthful diet, and lower obesity risk, prevalence, and incidence [13,14,15]. Similarly, physical activity, obesity and cardiometabolic disease may be influenced by public open space (POS; availability, proximity, size and quality), and walkable urban form (e.g., residential density and intersection density) [5,6,16,17].

Though there is a growing body of evidence linking built environment features to health behaviours and health outcomes, not all studies concur. Some studies have reported no association between the built environment (food stores or POS and walkable urban form) and diet, physical activity, or obesity [7,18,19,20]. Review articles have noted these inconsistent findings, ascribing them to differences in geographic regions, populations, and expressions of environmental measures (e.g., methods, definitions, source data, spatial scale used) [6,7,9]. To support evidence-based policy and actions aiming to improve population obesity, there is need for a far better understanding of the role of built environmental features and health.

Much of the research regarding environmental features and overweight/obesity is cross-sectional in design: more longitudinal studies are needed to support causal inference [6,9,16,21,22]. Cross-sectional studies limit inference on the direction of causation [6,23] and reverse causation is a possible explanation of any associations found. Longitudinal design allows for temporal antecedence to be established (i.e., that exposure precedes outcome, allowing for induction and latency), thus meeting one of the conditions necessary to indicate likelihood of causality [21,24]. Moreover, relatively few studies report simultaneously assessing multiple environmental factors (e.g., access/availability of POS and food resources, walkable urban form, greenness, and area SES) in relation to overweight/obesity [6,9]. Estimating the effects of singular built environment factors is analogous to assessing the influence of single individual-level risk factors on health outcomes. Such an approach ignores the interplay between the various built environment measures relating to health and precludes estimating the independent effects of built environment features in multivariable analyses. The aim of this study was to simultaneously assess associations between area SES, walkable urban form (dwelling density, intersection density, and land use mix), public open space (POS; count, area and greenness), food resources (fast food outlet count, and a relative food environment index) and 10-year change in body mass index (BMI) and waist circumference (WC).

## 2. Materials and Methods 

### 2.1. Study Sample and Region

This longitudinal study was part of the Place and Metabolic Syndrome (PAMS) Project and used three waves of North West Adelaide Health Study (NWAHS) data collected between 2000 and 2010. The NWAHS collected behavioural and clinical data for a population-based cohort of randomly selected adults (18 years and over), and was focused on the northwest region of Adelaide, a region which accounted for 38% of the city’s metropolitan area and 28% of the South Australian population in 2001 [25]. Written informed consent was provided by participants at each wave of data collection. Ethics approval for the PAMS Project was obtained from three Human Research Ethics Committees: University of South Australia (P029-10 and P030-10); Central Northern Adelaide Health Service (Queen Elizabeth Hospital; Application No. 2010010); and South Australian Department for Health and Ageing (Protocol No. 354/03/2013).

At Wave 1 (2000–2003), households within the north-west region of Adelaide were randomly sampled from the Electronic White Pages and the adult with the most recent birthday invited to participate in the study. Self-reported sociodemographic, behavioural and psychosocial information was collected using paper questionnaires and Computer Assisted Telephone Interviews. Participants were invited to attend a clinic visit at each data wave. Clinic visits were approximately 1 hour in duration and involved filling in the paper questionnaire (if not filled in prior) and collection by trained clinic staff of various measures including height, weight, and WC, blood pressure, and a 10mL fasting blood sample for assessment of triglycerides, cholesterol, and glycosylated haemoglobin (HbA1_c_) [26]. At Wave 1 (baseline), 4056 participants completed all modes of data collection. Cohort retention rates at Wave 2 (2004–2006; *n* = 3564) and Wave 3 (2008–2010; *n* = 2597) were 79% and 64%, relative to baseline. Further details of the data collection protocol are available elsewhere [26,27].

NWAHS participant residential addresses were geocoded and a geographic information system (GIS) used to derive baseline measures of neighbourhood built environments (i.e., walkable urban form, POS and greenness, and food resources) and sociodemographic composition (i.e., area SES) for each participant using ArcGIS (Version 9.3.1; ESRI, Redlands, CA, USA, 2010). The study region was restricted to urban areas defined as Census Collection Districts (CD) with a population density of at least 200 persons per square kilometre. The CD is the smallest spatial unit used by the Australian Bureau of Statistics to report Census data and includes approximately 225 dwellings [28].

### 2.2. Measures

#### 2.2.1. Outcome Measures—BMI and Waist Circumference

BMI (kg/m^2^) was calculated from height and weight measures. Height was measured without shoes to the nearest 0.5 cm using a wall-mounted stadiometer. Weight was measured without shoes, with the participant in light clothing, to the nearest 0.1 kg using standard digital scales. WC (nearest 0.1 cm, mean of three measures) was measured at the narrowest part of the waist, with the subject in a relaxed standing position, using an inelastic tape maintained in the horizontal plane.

#### 2.2.2. Built Environment

Measures were expressed for the study baseline. ArcGIS (Version 9.3.1; ESRI, Redlands, CA, USA, 2010) was used to calculate measures for 1600 metre (i.e., 1 mile) road-network distance buffers from participants’ residence (corresponding to 15–18 minute walk). The 1600 m buffer distance has previously been used in similar studies [9]. Source data for construction of measures came from the 2001 South Australian Digital Cadastral Database, Land Ownership and Tenure System database, 2001 Dwelling Count, 2001 Generalised Land Use and 1999 Retail Database from the Land Services Group, Department of Planning, Transport & Infrastructure, Government of South Australia, and 2001 Adelaide StreetPro© road data provided by Pitney Bowes Business Insight (Stamford, CT, USA).

The physical activity environment was represented by walkable urban form (intersection density, dwelling density, and land use mix) and POS (area, count, and greenness). Walkable urban form (“walkability”) was expressed using three measures: intersection density; dwelling density; and land use mix. Intersection density was defined as the count of intersections (i.e., nodes with travel directions greater than two) by NWAHS buffer area (intersection count/area of buffer intersected parcels in km^2^). Dwelling density was calculated as the count of dwellings by the total residential area within each NWAHS participant buffer (dwelling count/area of buffer intersected residential parcels in km^2^). An entropy index using residential, commercial, and recreational land use types represented land use mix [29]. These measures have previously been used to represent walkable urban form [20,29]. Higher scores are indicative of a more walkable urban form.

Public open space (POS) was expressed in three ways: area; count; and greenness. POS was defined as publicly-accessible land (≥150 square metres), including parks, gardens, wetlands, conservation reserves, sporting grounds, and recreation facilities (see Appendix A for a list of land use codes used). POS area was calculated as the total square metre area of all POS partially or completely contained within each participant buffer. POS count was defined as the number of POS partially or completely contained within each buffer. The Normalised Difference Vegetation Index (NDVI), an estimate of photo-synthetic plant activity based on Landsat satellite imagery (set at 25 metre grid cell resolution), was used to express “greenness” (median NDVI of buffer-intersected POS) [30]. Additional information on this method is available elsewhere [31].

Food environment measures included fast-food outlet count and the Retail Food Environment Index (RFEI). Food outlet information (location and type) was obtained from the 1999 South Australian Retail Database (Planning SA, South Australian Government, Adelaide, SA, Australia) and outlets were classified as either healthful (greengrocers, butchers, health food stores, supermarkets and healthful take-away), fast food (chain or non-chain fast-food outlets), or unhealthful retail (bakeries, sweet food retailers, convenience stores, and service station food-marts) as per classifications designed by a dietician for use in a previous Australian study [32]. Details of food outlet classifications are provided in Appendix B. 

Fast-food outlet count was defined as total count of fast-food outlets (chain and non-chain) within each buffer. The RFEI, a measure of the relative unhealthfulness of the food environment, was calculated as the ratio of unhealthful to healthful food outlets (n unhealthful outlets/ (n healthful outlets +1)) within each buffer [33]. 

#### 2.2.3. Area Socioeconomic Status

Area SES was expressed using area education defined as the proportion of individuals who were university educated (bachelor degree or higher) residing within the State Suburb. State Suburbs are formed by aggregating CDs to align with the most recent gazetted suburb at the time of the Census [34]. State Suburbs for 2001 were used to align with Wave 1 of the NWAHS. Though composite indices of area SES are commonly used, their usage prevents inference regarding specific area-level SES effects [35]. Further, composite indices are country-specific reducing comparability between studies from different countries [23]. Education is commonly used to indicate SES in epidemiological studies as it is usually established relatively early in life and does not vary as can income, employment or occupation.

#### 2.2.4. Covariate Variables 

Individual-level covariates included baseline participant age, sex (male (reference) or female), marital status (married/de facto or not married/de facto (reference)), smoking status (smoker or non-smoker (reference)), employment status (full/part time employed (reference) or not currently employed), and education (university educated (degree or higher) or not (reference)). These measures were selected due to their relationship with NWAHS cohort attrition and data missingness, and potential for confounding of estimated relationships. Their inclusion in models satisfies the analytic criterion of missing at random [36]. 

### 2.3. Data Analysis 

Descriptive statistics including correlations were computed (SAS, version 9.4, SAS Institute Inc., Cary, NC, USA) for baseline values of socio-demographic, BMI and WC, and environmental measures.

Associations between baseline environmental exposures and changes in outcomes (BMI or WC) across the 10-year follow up were estimated using latent variable growth models in Mplus (version 8.3, Muthen & Muthen, Los Angeles, CA, USA, 1998-2017). Change in BMI or WC was modelled using latent variables (slope for change over time, and intercept for baseline) with random effects to allow for participant-specific variations. For both BMI and WC (separately), relationships between environmental exposures and covariates, and latent variables for intercept (i.e., baseline) and slope (i.e., change) were estimated. As there were only three waves of data, only linear growth curves were considered [37]. Models accounted for clustering of participants within suburbs. 

The analytic sample was smaller than the initially available Wave 1 sample (n = 4056) for the following reasons: lack of valid (geo-codable) residential address (n = 15); residence outside of the urban area (n = 154) (applied as associations between environmental exposures and health outcomes may vary between urban and rural regions [38]); residing in a suburb with less than four other participants (n = 24) (applied to allow adequate estimation of within suburb effects for model adjustment of suburb-level participant clustering); and incomplete data at Wave 1 (n = 147). 551 participants who moved between Wave 1 and Wave 2 were excluded due to the consequent mismatch between baseline environmental measures and actual environmental exposures. Participants who moved after Wave 2 (n = 241) were retained in the analytic sample as the majority of their exposure period was to the environmental measures included as predictors (latency effect). Lastly, on average, the NWAHS cohort had worsening health trajectories, aligning with expectations based on associations between aging and normative, age-related increases in BMI. Consequently, to reduce the influence of a potential ceiling effect, the analytic sample was restricted to cohort participants who were not obese at baseline (BMI < 30 kg/m^2^), which resulted in the further loss of 912 participants, leaving an analytic sample of 2253.

Analytic models used the full information maximum likelihood (FIML) approach to account for incomplete records. FIML deals directly with missing data yielding asymptotically efficient estimates under the assumed model, with standard errors of estimates accounting for the incomplete nature of the data [39]. Area-level measures (environmental exposures) were standardised (i.e., z-scores with a mean of 0 and standard deviation of 1) prior to inclusion in models to allow for comparison of the size of estimated effects. Akaike’s Information Criteria (AIC) values are provided to indicate difference in model fit between models. Statistical significance level was alpha of 5%. 

## 3. Results

Information on the baseline sample and their environmental exposures is presented in Table 1. Mean participant age was 51 years with similar proportions of women (50.8%) and men (49.2%) in the sample. 12.2% of the sample were university educated though mean area-level (university) education was 8.1%. Mean POS count was 23 parcels, mean fast food count was 7 outlets, and the mean RFEI was 2 indicating a greater availability of unhealthful relative to healthful food resources. 

Correlations between baseline participant BMI, waist circumference, and environmental measures are presented in Table 2. Baseline BMI was statistically significantly correlated with area SES (Spearman’s rho −0.04, *p* < 0.05) but no other area exposure measure. Baseline WC in men was not statistically significantly correlated with any environmental measure though WC in women was statistically significantly correlated with area SES (rho −0.10, *p* < 0.01), dwelling density (rho −0.07, *p* < 0.05), and RFEI (rho 0.07, *p* < 0.05).

Most but not all environmental measures were statistically significantly correlated. Notable moderate to large correlations (in descending order of magnitude) were: area SES with dwelling density (Spearman’s rho 0.53, *p* < 0.0001); intersection density with count of fast food outlets (rho 0.45, *p* < 0.0001); POS count with POS area (rho 0.38, *p* < 0.0001); dwelling density with intersection density (rho 0.37, *p* < 0.0001); and dwelling density with count of fast food outlets (rho 0.35, *p* < 0.0001). 

Small to moderate correlations included: area SES with fast food count (rho 0.28, *p* < 0.0001), greenness (rho 0.28, *p* < 0.0001), and intersection density (rho 0.26, *p* < 0.0001); land use mix negatively correlated with intersection density (−0.20, *p* < 0.0001); POS area positively correlated with land use mix (0.23, *p* < 0.0001) but negatively with intersection density (rho −0.22, *p* < 0.0001); and NDVI positively correlated with count of fast food outlets (0.25, *p* < 0.0001) and intersection density (rho 0.20, *p* < 0.0001), and negatively correlated with POS count (rho −0.13, *p* < 0.0001). Due to the level of correlation and the potential for collinearity issues in inferential models, variance inflation factors (VIF) were assessed using simple regression models in SAS (version 9.4, SAS Institute Inc., Cary, NC, USA). As complex models in SAS and models in Mplus do not provide calculation of VIFs, simple models were used. Separate models included all predictors of interest in relation to wave 1 BMI and change in BMI (wave 1 to wave 2). All VIFs were less than 2 and therefore collinearity was not considered to be an issue.

Intraclass correlations (ICCs) could not be directly calculated from the Mplus latent growth models so were calculated from covariance parameter estimates obtained from a three-level random effects model with no predictors performed in SAS (version 9.4, SAS Institute Inc., Cary, NC, USA) [40]. Estimated ICCs indicated high correlation within individuals (repeated BMI or waist measures over time: ICC _BMI_ 0.84; ICC _waist_ 0.81) and very low correlation within suburbs (i.e., area clustering: ICC _BMI_ 0.002; ICC _waist_ 0.003).

Analytic models estimating associations between environmental exposures and change in BMI or WC began with a model including only area SES and individual covariates (Model 1) which was sequentially developed to include all environmental measures of interest (Model 4). Model fit, based on AIC, worsened for each set of models (BMI or WC) as more measures were included in the models suggesting Model 1 had the best fit. This was most notable for BMI.

On average, BMI increased by 0.43 kg/m^2^ per year (*p* < 0.0001). This was consistent across all models. WC was estimated to increase at 0.94 to 0.96 cm per year across models (*p* < 0.0001).

Area SES was not statistically significantly associated with change in BMI in the model including only itself and individual covariates (see Table 3), however, on addition of other environmental measures area SES became consistently negatively associated with increasing BMI (i.e., a protective effect; estimate ranging from −0.024 to −0.025 kg.m^2^ across Models 2–4 *p* < 0.05). 

Of the measures representing walkable urban form, only dwelling density was statistically significantly positively associated with increasing BMI (estimate 0.019 (95%CI 0.004, 0.035) *p* < 0.05, Model 2). This association was slightly strengthened with the addition of other environmental measures (estimate 0.022 (95%CI 0.008, 0.037), *p* < 0.01, Model 3–4).POS area was positively associated with increasing BMI (estimate 0.019 (95%CI 0.001, 0.037), *p* < 0.05, Model 3). This effect was not robust to the inclusion of food environment measures (Model 4). POS count and greenness were not associated with change in BMI. No food environment measures were associated with change in BMI (Model 4).

The pattern of associations between change in WC and environmental exposures was similar to that for models predicting change in BMI (Table 4). Area SES was negatively associated with increasing WC (protective effect) once other environmental predictors were included in the models (estimates ranged from −0.083 in Model 2 to −0.098 in Model 4, *p* < 0.01). Dwelling density was consistently positively associated with increasing WC with estimates ranging from 0.093 in Model 2 to 0.108 in Model 3 (*p* < 0.0001). POS area was also positively associated with increasing WC (estimate 0.063 (95% CI 0.017,0.110), *p* < 0.01, Model 3), however, unlike in the BMI models, this effect remained statistically significant on inclusion of food environment measures. Food environment measures were not associated with change in WC.

## 4. Discussion

Few studies simultaneously assess multiple residential environment features in relation to overweight/obesity. In this study, when modelling multiple built environment measures, only area SES (negatively), dwelling density and POS area (both positively) were associated with 10-year change in BMI and WC (hereafter, collectively: body size). In our multivariable models, other components of walkable urban form (intersection density and land use mix), POS count, greenness, and measures of the food environment were not related to rate of change in body size.

This study’s findings that greater area SES had a protective health affect aligns with previous cross-sectional and longitudinal studies [6,7]. Area SES has been well studied in relation to health outcomes including body size, is well accepted as being important in relation to health and will therefore not be discussed at length in this paper.

This study considered components of walkable urban form rather than walkability as an index. The unexpected finding that greater dwelling density was associated with increasing body size over time highlights the importance of considering ‘walkability’ factors separately. Though estimates of associations between the other walkability components (i.e., intersection density and land use mix) and change in body size do not reach statistical significance, their beta coefficients are in the negative direction, that is, the opposite of that for dwelling density. In additional models performed using walkability as an index (sum of deciles for the three included factors, results not shown), the beta coefficient for walkability was not statistically significant. This suggests that the common practice [41,42] of using of a composite walkability index may obscure the true drivers of associations with change in body size. Additionally, consideration of individual walkability components may be more relevant to health policy and practice, providing more specific insights to inform interventions. 

It is generally assumed that greater dwelling density is supportive of local walking participation due to the increased likelihood of local destinations. Our findings partially support this notion; indeed, greater dwelling density was moderately correlated with greater number of nearby fast-food outlets which, although potential destinations, are unfortunately often associated with increased prevalence of obesity [14,43]. Though dense areas may promote travel-related walking behaviour, less dense areas promote leisure walking, with the result that total physical activity is not necessarily different [44]. High dwelling density may also have adverse mental and emotional health consequences [44], especially in poorly designed areas with high noise-levels and traffic exposure [45], while poorer mental health can increase risk of developing obesity [46,47,48]. Consequently, relationships between dwelling density and body size may not be simple.

Of other built environment measures assessed in this study, only POS area was associated with increasing body size, particularly WC. Review articles have concluded that POS/greenness is largely inversely associated with obesity, BMI and CVD risk e.g., [6,49] but that these associations are inconsistent. The diversity of strategies used to conceptualise and operationalise POS/greenness has been proposed as an explanation for this inconsistency [50]. This premise is supported by our findings and those of a recent Dutch cross-sectional study which assessed three greenspace measures in relation to physical activity and overweight: 1) road-distance to park entrance; 2) mean NDVI within circular buffers; and 3) proportion of greenspace within a circular buffer [51]. The authors reported mean NDVI had the strongest associations with outdoor physical activity and (inversely) with obesity. In contrast, our study found that POS area was associated with worsening body size while POS count and NDVI were not. Our findings, together with the recent work in The Netherlands [51], exemplify the need to carefully approach the conceptualisation and operationalisation of built environment measures using clear hypotheses for the mechanism of linkage to health outcomes. It is generally assumed that greater POS area enables physical activity, however, greater POS may also inhibit local walking for basic tasks as greater POS area may increase distances to local resources. This reflects the physical complexity of residential areas and the multiple relationships between environmental features.

All environmental measures in this study were correlated, though some correlations were expected, for example, between expressions of similar measures (e.g., POS area and count (rho 0.579, *p* < 0.0001)). Of note, area SES was significantly correlated with all built environment measures except POS area and RFEI. These multiple correlations highlight the potential for confounding in studies that assess only a limited number of built environment measures. Though the inclusion of area SES in analytic models to account for potential confounding is supported [52], residual confounding may remain and partly account for the inconsistent findings reported in the literature. Moreover, the extent of correlations observed here highlights the importance of considering multiple environmental measures simultaneously in models to identify which environmental features to prioritise in intervention strategies. 

Our findings suggest that the residential physical activity environment may have greater impact on change in body size than the food environment. However, while POS area and dwelling density were, in this study, the only built environment measures related to change in body size, we do not discount a plausible role for other features that could support healthful or unhealthful behavioural patterns, including diet and physical activity [21]. Built environment factors may have only a small impact on behaviour, an effect that may be insufficient to substantially impact change in body size given other individual and environmental factors, particularly over relatively short time periods. The cumulative combination of such effects, however, may have a significant impact on body size over the life-course [53]. In addition, the availability of food resources, such as fast food outlets, is thought to function as an enabler supporting unhealthful dietary behaviour. Whilst there was variation in food resource availability within the study region, it is possible that participants had more than adequate availability of outlets to easily obtain desired foods thus resulting in null association between the food environment and change in body size.

This study addressed several methodological limitations observed in previous studies. It was longitudinal in design with environmental exposure measures preceding the outcome, thus supporting causal inference through temporality of measures [22]. Change in body size was assessed using clinical measures of BMI and WC, avoiding self-report bias. 

Another strength is that built environment measures were defined within road-network buffers centred on participant residential address, as recommended previously [54]. The 1600m buffer distance is typically considered ‘walkable’ by adults in 15–20 minutes, likely to capture local walk-accessible destinations, is commonly used in the Australian context [55,56,57] and thus supports comparison with previous studies. However, road-network buffers do not typically account for potential local barriers such as crossings of busy arterial roads nor do they necessarily align with what an individual may perceive as their local area. 

Area SES was expressed within predefined administrative units (suburbs) as these data were only available in this format. Use of data aggregated within pre-defined spatial units has received criticism due to the potential influence of the Modifiable Areal Unit Problem where differences in the size (scale) and shape (zonation) of spatial units results in difference statistical associations [58]. Given, however, the consistency of findings regarding area SES and obesity across multiple studies in different regions, and using different aggregations of administrative units, there is little doubt regarding the adverse effect of low area SES on health.

Multiple measures of the built environment were included in analytic models, however, these measures do not represent all potential built environment influences on behaviour and health. For example, our components of walkable urban form do not include other environmental aspects such as surface condition or route aesthetics that may relate to walking behaviour [59]. Similarly, this study considers POS in terms of count, area, and greenness; it does not include other features relating to POS quality that may influence behaviour and health outcomes [60]. The environmental characteristics assessed may have changed during the cohort follow-up period, however there was little evidence of area gentrification during this time period [61]. Though this study concurrently assessed multiple residential environment features in relation to overweight/obesity, it did not assess any interactions effects; the influence of environmental characteristics may be modified by other environmental characteristics or by individual-level factors. Lastly, environments other than an individual’s residential environment likely impact on health outcomes, for example, an individual’s work environment.

## 5. Conclusions

This study concurrently assessed the independent effects of multiple built environment features and area SES on 10-year change in body size. In this region and sample, area SES (education) was protective against the development of increasing body size, independent of other built environmental measures. Greater dwelling density and POS area exacerbated rates of increasing body size. Few studies have simultaneously assessed multiple measures of residential environments in relation to body size making this an important contribution to the literature.

The findings of this study align with previous research indicating that area SES is a key factor predicting the evolution of adverse health outcomes for populations so exposed. However, unexpected correlations between built environment measures suggest that confounding may be an issue within studies that simultaneously assess few built environment features in models, especially when not accounting for area SES. Area SES and the physical activity-related environment appeared in our study more important than the food environment in relation to body size, however this may be due to the generally substantial availability of food outlets throughout the study region.

## Figures and Tables

**Table 1 ijerph-17-00870-t001:** Sociodemographic characteristics of the sample (baseline) and environmental exposures.

**Individual-Level Characteristics**	**n**	**Mean or %**	**SD**	**Median**	**Min**	**Max**
Age (Years)	2253	51.0	16.9	50.0	18.0	90.0
Sex: Male	1109	49.2%	-	-	-	-
Female	1144	50.8%	-	-	-	-
Marital Status: married/de facto	1390	61.7%	-	-	-	-
not married/ de facto	863	38.3%	-	-	-	-
Smoking Status: smoker	442	19.6%	-	-	-	-
non-smoker	1811	80.4%	-	-	-	-
Employment Status: full/part-time employed	1200	53.3%	-	-	-	-
not currently employed	1053	46.7%	-	-	-	-
Education: university educated	275	12.2%	-	-	-	-
not university educated	1978	87.8%	-	-	-	-
**Environmental exposures ^a^**	**n**	**Mean or %**	**SD**	**Median**	**Min**	**Max**
Area SES	2253	8.1%	4.74	6.65	0.92	21.24
Dwelling density (n/km^2^)	2253	1583.74	311.73	1577.47	83.67	2705.17
Intersection density (n/km^2^)	2253	44.59	21.01	46.68	2.50	91.95
Land use mix	2253	0.64	0.14	0.64	0.08	1.00
POS Area (total km^2^)	2253	1.00	0.67	0.94	0.04	4.64
POS count	2253	23.4	9.3	22.0	5.0	55.0
Greenness (median NDVI)	2253	−4.3	6.1	−5.0	−20.0	17.0
Fast food count (n)	2253	7.2	4.5	7.0	0.0	26.0
RFEI	2253	2.3	1.5	2.0	0.0	13.0

^a^ Area SES (proportion with a University Degree or higher) expressed for the State Suburb; all other environmental variables expressed for 1600m residence-centred road-network buffers. Abbreviations: SES: socioeconomic status; POS: Public Open Space; NDVI: Normalised Difference Vegetation Index; RFEI: Retail Food Environment Index.

**Table 2 ijerph-17-00870-t002:** Spearman rank correlation coefficients between baseline BMI, WC (by sex) and environmental exposures.

		1	2	3	4	5	6	7	8	9	10	11	12
1	Baseline BMI	1 n = 2253	-	-	-	-	-	-	-	-	-	-	-
2	Baseline WC (males)	-	1 n = 1109	-	-	-	-	-	-	-	-	-	-
3	Baseline WC (females)	-	-	1 n = 1144	-	-	-	-	-	-	-	-	-
4	Area SES	−0.043* n = 2253	−0.023 n = 1109	−0.097** n = 1144	1 n = 2253	-	-	-	-	-	-	-	-
5	Dwelling Density (per km^2^ within 1600m buffer)	−0.033 n = 2253	−0.005 n = 1109	−0.074* n = 1144	0.528**** n = 2253	1 n = 2253	-	-	-	-	-	-	-
6	Land use mix (Entropy value within 1600m buffer)	0.016 n = 2253	0.013 n = 1109	0.031 n = 1144	−0.056** n = 2253	0.076*** n = 2253	1 n = 2253	-	-	-	-	-	-
7	Intersection Density (per km^2^ within 1600m buffer)	−0.023 n = 2253	−0.057 n = 1109	−0.030 n = 1144	0.261**** n = 2253	0.367**** n = 2253	−0.203**** n = 2253	1 n = 2253	-	-	-	-	-
8	POS Count (number of POS parcels within 1600m buffer)	0.005 n = 2253	−0.013 n = 1109	0.045 n = 1144	−0.161**** n = 2253	−0.085**** n = 2253	0.031 n = 2253	0.006 n = 2253	1 n = 2253	-	-	-	-
9	POS Area-all (Total m^2^ area per buffer)	−0.021 n = 2253	0.011 n = 1109	0.024 n = 1144	−0.021 n = 2253	−0.183**** n = 2253	0.234**** n = 2253	−0.224**** n = 2253	0.375**** n = 2253	1 n = 2253	-	-	-
10	Greenness (Median NDVI per all POS within buffer)	<−0.001 n = 2253	−0.035 n = 1109	−0.013 n = 1144	0.281**** n = 2253	0.150**** n = 2253	−0.015 n = 2253	0.199**** n = 2253	−0.132**** n = 2253	−0.074*** n = 2253	1 n = 2253	-	-
11	Fast Food Count (Fast Food within buffer)	−0.002 n = 2253	0.013 n = 1109	−0.015 n = 1144	0.283**** n = 2253	0.346**** n = 2253	−0.014 n = 2253	0.446**** n = 2253	−0.024 n = 2253	−0.155**** n = 2253	0.248**** n = 2253	1 n = 2253	-
12	Retail Food Environment Index (RFEI per buffer)	−0.004 n = 2253	−0.054 n = 1109	0.071* n = 1144	−0.029 n = 2253	0.063** n = 2253	0.106**** n = 2253	0.035 n = 2253	0.0106 n = 2253	−0.030 n = 2253	−0.240 n = 2253	0.258**** n = 2253	1 n = 2253

^a^ Age and BMI measured at the individual-level, Area SES was expressed for the State Suburb, all other environmental variables were expressed for 1600m residence-centred road-network buffers. Abbreviations: BMI: Body Mass Index; WC: Waist Circumference; SES: socioeconomic status; POS: Public Open Space; NDVI: Normalised Difference Vegetation Index; RFEI: Retail Food Environment Index. * *p* < 0.05; ** *p* < 0.01; *** *p* < 0.001; **** *p* < 0.0001.

**Table 3 ijerph-17-00870-t003:** Estimates of associations between environmental exposures and change in BMI (n = 2253, suburbs n = 121).

	Model I AIC = 23,344.67	Model II AIC = 23,350.06	Model III AIC = 23,354.58	Model IV AIC = 23,361.57
	Estimate (95% CI)	Estimate (95% CI)	Estimate (95% CI)	Estimate (95% CI)
Age	−0.007**** (−0.008, −0.005)	−0.007**** (−0.008, −0.005)	−0.007**** (−0.008, −0.005)	−0.007**** (−0.008, −0.005)
Sex (female)	0.056**** (0.030, 0.083)	0.056**** (0.030, 0.082)	0.056**** (0.030, 0.082)	0.055**** (0.029, 0.081)
Employment (not employed)	0.028 (−0.011, 0.067)	0.028 (−0.011, 0.068)	0.029 (−0.011, 0.068)	0.028 (−0.012, 0.068)
Education	−0.035 (−0.075, 0.006)	−0.034 (−0.075, 0.006)	−0.033 (−0.073, 0.007)	−0.033 (−0.073, 0.007)
Marital Status	−0.016 (−0.048, 0.017)	−0.013 (−0.045, 0.019)	−0.014 (−0.047, 0.018)	−0.014 (−0.046, 0.018)
Smoking Status (smoker)	0.081** (0.035, 0.128)	0.081** (0.035, 0.128)	0.080** (0.034, 0.127)	0.081** (0.034, 0.127)
Area SES	−0.016 (−0.034, 0.001)	−0.024* (−0.042, −0.005)	−0.025** (−0.043, −0.007)	−0.025* (−0.043, −0.006)
Dwelling Density	-	0.019* (0.004, 0.035)	0.022** (0.008, 0.037)	0.022** (0.008, 0.037)
Intersection Density	-	−0.005 (−0.022, 0.011)	−0.003 (−0.019, 0.014)	−0.003 (−0.021, 0.014)
Land use mix	-	−0.005 (−0.021, 0.012)	−0.009 (−0.026, 0.008)	−0.010 (−0.027, 0.008)
POS area	-	-	0.019* (0.001, 0.037)	0.018 (−0.001, 0.037)
POS count	-	-	0.001 (−0.017, 0.019)	0.001 (−0.017, 0.019)
Greenness (NDVI)	-	-	−0.003 (−0.018, 0.013)	−0.003 (−0.018, 0.013)
Fast food count (n)	-	-	-	−0.001 (−0.016, 0.014)
RFEI	-	-	-	0.006 (−0.012, 0.024)

Area SES was expressed for the State Suburb, all other environmental variables were expressed for 1600m residence-centred road-network buffers. Abbreviations: BMI: Body Mass Index; SES: socioeconomic status; POS: Public Open Space; NDVI: Normalised Difference Vegetation Index; RFEI: Retail Food Environment Index. **** *p*-value <0.0001; *** *p*-value <0.001; ** *p*-value <0.01; * *p*-value <0.05.

**Table 4 ijerph-17-00870-t004:** Estimates of associations between environmental exposures and change in WC (n = 2253, suburbs n = 121).

	Model I AIC = 35,905.55	Model II AIC = 35,907.31	Model III AIC = 35,908.26	Model IV AIC = 35,914.08
	Estimate (95% CI)	Estimate (95% CI)	Estimate (95% CI)	Estimate (95% CI)
Age	−0.013**** (−0.017, −0.008)	−0.013**** (−0.018, −0.008)	−0.013**** (−0.018, −0.009)	−0.013**** (−0.018, −0.009)
Sex (female)	0.115* (0.020, 0.209)	0.112* (0.018, 0.206)	0.110* (0.016, 0.205)	0.110* (0.015, 0.205)
Employment (not employed)	0.122 (−0.003, 0.247)	0.124 (−0.001, 0.249)	0.126 (0.000, 0.251)	0.123 (−0.003, 0.250)
Education	−0.099 (−0.227, 0.029)	−0.097 (−0.226, 0.031)	−0.097 (−0.224, 0.030)	−0.101 (−0.228, 0.026)
Marital Status	−0.032 (−0.146, 0.082)	−0.019 (−0.133, 0.094)	−0.022 (−0.136, 0.091)	−0.020 (−0.134, 0.094)
Smoking status (smoker)	0.313**** (0.191, 0.434)	0.314**** (0.194, 0.433)	0.314**** (0.194, 0.434)	0.313**** (0.192, 0.433)
Area SES	−0.049 (−0.099, 0.000)	−0.083** (−0.134, −0.032)	−0.096**** (−0.145, −0.047)	−0.098**** (−0.148, −0.048)
Dwelling Density	-	0.093**** (0.043, 0.143)	0.108**** (0.057, 0.159)	0.106**** (0.057, 0.155)
Intersection Density	-	−0.033 (−0.087, 0.020)	−0.028 (−0.079, 0.023)	−0.035 (−0.089, 0.019)
Land use mix	-	−0.025 (−0.075, 0.025)	−0.040 (−0.089, 0.010)	−0.042 (−0.091, 0.006)
POS area	-	-	0.063** (0.017, 0.110)	0.064* (0.015, 0.114)
POS Count	-	-	−0.026 (−0.082, 0.031)	−0.026 (−0.082, 0.030)
Greenness (NDVI)	-	-	0.012 (−0.038, 0.062)	0.010 (−0.040, 0.060)
Fast Food Count (n)	-	-	-	0.019 (−0.036, 0.073)
RFEI	-	-	-	0.013 (−0.038, 0.065)

Area SES was expressed for the State Suburb, all other environmental variables were expressed for 1600m residence-centred road-network buffers. Abbreviations: WC: Waist Circumference; SES: socioeconomic status; POS: Public Open Space; NDVI: Normalised Difference Vegetation Index; RFEI: Retail Food Environment Index. **** *p*-value <0.0001; *** *p*-value <0.001; ** *p*-value <0.01; * *p*-value <0.05.

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
