# Peer review of "Contributions of Multiple Built Environment Features to 10-Year Change in Body Mass Index and Waist Circumference in a South Australian Middle-Aged Cohort"

_ijerph, 2020, doi:10.3390/ijerph17030870_

Round 1
Reviewer 1 Report
This is a very nice and urgently needed study as it is one of the few longitudinal studies to examine the association of the built environment and weight outcomes. The paper would have been even stronger if the authors had examined changes to the built environment over the ten year study period.
Were participants asked how long they had lived at their residence at baseline and/or are the authors aware of what the built environment factors were like in the years before baseline data was collected? In the absence of accounting for changes over the course of the study, it seems important to be aware of those types of changes in the preceding years.
Although the authors mention that clinic visits were previously described, please briefly describe them here too.
Please include a table to includes how each food outlet type is defined.
Reviewer 2 Report
Thank you for the opportunity to review this manuscript on the relation between multiple built environment features and changes in body mass index and waist circumference over a 10 year period. With the 10-year follow-up time and thorough description of the study procedures, this manuscript is an addition to the field.
As already suggested to the editors, I would suggest to include this manuscript in IJERPH’s special issue on Upstream determinants of lifestyle behaviors and chronic diseases (https://www.mdpi.com/journal/ijerph/special_issues/lifestyle_behaviors) which has just opened its submissions.
I would like the authors to comment on their longitudinal design; please clarify how exactly a longitudinal design contributes to causal inference, and why the authors did not investigate the relation between changes in built environment features and changes in the outcome. Related to these questions, please also explain why people who moved between wave 1 and 2 (but not if they moved between wave 2 and 3) were excluded.
Minor comments:
Include info on age range and sampling area in abstract L33: Unclear what is meant with ‘excess body weight as a function of excess adiposity’. L70: would be good to clarify how longitudinal designs contirbute to causal inference. L142-147: were other food outlet types (e.g., full service restaurants) not included in the RFEI? And how was the RFEI calculated when there were nu healthful outlets (which means your numerator is zero, and you can’t divide zero by a denominator). L148: how were fast-food outlets defined? L183-186 Please describe both reasons and numbers of excluded participants L191: did the area-level measures allow for standardisation? I.e., were they normally distributed? Why were no interactions between environmental factors examined? Table 1: you could consider presenting the total POS area as square kilometers instead of square meters. Line 246-266: it would be helpful to describe associations as positive (+) or negative (-), I believe that is more intuitive than describing associations with ‘worsening BMI’. Table 3: the table suggests than BMI changes is a determinant of the outcome (change in BMI). What is the estimate provided here? A constant? This should be made clearer. Discussion: I would have expected a paragraph describing the area deprivation effects on body size. Why is this not discussed? Line 287-297: I am not sure I agree with this rationale. The idea behind a walkability index is that the combination of components has an effect on behaviours or health outcomes. That is, dwelling density combined with intersection density combined with destinations, etc. While dwelling density itself was associated with body size in an unexpected direction, a walkability index may still have shown results in the expected direction. Please clarify better what the advantages are of using single components, and how these results relate to the results from studies that used indexes. Line 299: I would suggest to simply write: ‘larger POS areas were associated with increasing body size’, rather than ‘was associated with change (worsening effect)’. Line 339: I think it is a bit too simple to refer to one French study to back up the decision to use 1600m road-network buffers. In my opinion, buffer sizes should be dependent on the local context. I would like to see a stronger rationale for why it is a strength to use 1600m road-network buffers in this Australian context. Line 367: Is this the main explanation that the authors have fort heir null-findings? If so, please state that clearly early on in the discussion section. However, I tend to disagree that a 10-year period is not long enough to detect effects on body size. There could be a lot of other explanations, including the potential misclassification of exposure due to the buffer around residential areas, not including movers and thus limiting variability/power, and the fact some factors simply are not influencing body size. Please discuss the influence of removing movers from the analytical sample. Line 374: I disagree with the proposition that area SES and the physical-activity environment are more important than the food environment. The authors used six PA environment factors and 2 food environment factors, and found only one PA environment factor to be associated with both BMI and WC. I think it is not helpful in this field to dichotomize or single out factors, especially given the finding that dwelling density was quite strongly correlated with fast food counts. I therefore think the final conclusions warrant a bit more nuance.Author Response
Please see the attachment
